# Bone Microenvironment and Osteosarcoma Metastasis

**DOI:** 10.3390/ijms21196985

**Published:** 2020-09-23

**Authors:** Chaofei Yang, Ye Tian, Fan Zhao, Zhihao Chen, Peihong Su, Yu Li, Airong Qian

**Affiliations:** 1Lab for Bone Metabolism, Key Lab for Space Biosciences and Biotechnology, School of Life Sciences, Northwestern Polytechnical University, Xi’an 710072, China; yangchaofei@mail.nwpu.edu.cn (C.Y.); tianye@nwpu.edu.cn (Y.T.); sofan@mail.nwpu.edu.cn (F.Z.); chzhh@mail.nwpu.edu.cn (Z.C.); suph@mail.nwpu.edu.cn (P.S.); liyu@nwpu.edu.cn (Y.L.); 2Research Center for Special Medicine and Health Systems Engineering, School of Life Sciences, Northwestern Polytechnical University, Xi’an 710072, China; 3NPU-UAB Joint Laboratory for Bone Metabolism, School of Life Sciences, Northwestern Polytechnical University, Xi’an 710072, China

**Keywords:** bone microenvironment, metastasis, osteosarcoma, primary bone tumor, signal pathway

## Abstract

The bone microenvironment is an ideal fertile soil for both primary and secondary tumors to seed. The occurrence and development of osteosarcoma, as a primary bone tumor, is closely related to the bone microenvironment. Especially, the metastasis of osteosarcoma is the remaining challenge of therapy and poor prognosis. Increasing evidence focuses on the relationship between the bone microenvironment and osteosarcoma metastasis. Many elements exist in the bone microenvironment, such as acids, hypoxia, and chemokines, which have been verified to affect the progression and malignance of osteosarcoma through various signaling pathways. We thoroughly summarized all these regulators in the bone microenvironment and the transmission cascades, accordingly, attempting to furnish hints for inhibiting osteosarcoma metastasis via the amelioration of the bone microenvironment. In addition, analysis of the cross-talk between the bone microenvironment and osteosarcoma will help us to deeply understand the development of osteosarcoma. The cellular and molecular protagonists presented in the bone microenvironment promoting osteosarcoma metastasis will accelerate the exploration of novel therapeutic strategies towards osteosarcoma.

## 1. Introduction

The bone microenvironment is composed of bone marrow and a mineralized extracellular matrix. Bone marrow contains two different cell types: hematopoietic stem cells with hematopoietic function and bone marrow mesenchymal stem cells, which are responsible for differentiating into non-blood cells components in bone including mesenchymal stem cells (MSCs), osteoblasts, osteoclasts, osteocytes, fibroblasts, fat cells, etc. [1,2,3]. The bone microenvironment provides an ideal site for many distinct cancers to thrive. For example, some secondary tumors which originate in other sites tend to diffuse into the bone including breast cancer, prostate cancer, etc. [4,5,6]. The concept of the “vicious cycle” has been present since the 1990’s to describe this close connection between metastatic tumor cells and bone cells [7]. Tumor cells invade bone, resulting in the loss of balance between bone formation and bone resorption [4,5].

The bone microenvironment is also a fertile soil and a complex biological system that facilitates the metastasis of many cancers including osteosarcoma (OS) [8,9]. In recent years, to understand the progress of OS, a number of studies have focused on the relationship between OS and the bone microenvironment, for example, MSCs, which are one of the most important members in the bone microenvironment [10]. Here, we summarized the research approaches toward the bone microenvironment and OS metastasis. We reviewed the promoting effect of the main components in the bone microenvironment on OS metastasis such as mesenchymal stem cells, hypoxia, acidosis, and chemokine. Meanwhile, we were particularly concerned with the signaling pathways that were activated by these factors in the bone microenvironment and the consequences for the progress of OS metastasis.

OS is the main malignant primary bone tumor in young adults and children. The metaphysis regions of long bones, the most active sites in bone, are major sites of OS growth [11]. The large tumor heterogeneity in OS creates some difficult issues for tumor therapy such as identification of reliable biomarkers, recognizing the mechanism of recurrence, and identifying which cell type causes OS [12].The conventional form of OS can be broken down to distinctive subtypes based on histological analysis: osteoblastic, fibroblastic, chondroblastic, epithelioid, giant-cell rich, small-cell and telangiectatic types [13]. Although the morphological characteristics are different, the mainstream research believes that OS emerges in osteoids generated from mesenchymal stem cells as well as pre-osteoblasts or osteoblast precursors [14]. For example, some papers reported that the deletion of *TP53* and *Rb* can cause OS transformation of osteoblasts [15,16,17]. The loss of *Rb* can trigger the transformation of MSCs into OS, and the overexpression of C-myc also has similar consequences for MSCs [18]. The molecular pathogenesis of OS is complicated, but despite this, there are some key genes that have been studied and can give us a little inspiration. The *TP53* and *Rb* are the genes with the highest frequencies of absence and mutations in human OS and transgenic mouse models. Both *TP53* and *Rb* are the tumor suppressor genes and involved in cell cycle regulation, while *TP53* is involved in cell apoptosis. There are other genes with aberrant expression in OS, including c-myc, AP-1, c-fos, TWIST, MMP, IGF-1, etc. [19,20,21,22].

Tumor metastasis is the primary problem for tumor therapy [23]. Most OS infiltrate the surrounding tissue, and even metastasize to the lung when they are found. Lung metastasis is the main challenge for OS therapy; however, due to the introduction of chemotherapy, the five-year survival rate of OS has increased to about 70% since the 1970s, but the five-year survival rate still remains as low as 20–30% after lung metastasis [24]. The study of how the spread and metastasis of OS is affected by the tumor microenvironment is not yet in-depth. Malignant OS cells form a complex mixture with other normal cells (MSCs, fibroblasts, osteoblasts, and myeloid immune cells) and some chemical factors (hypoxia, acidosis). This special tumor microenvironment of the complex mixture is a perfect place for OS to develop and metastasize. Understanding this special bone microenvironment, the mechanism of OS metastasis can be better understood, and it is possible to find a therapeutic target for the treatment of OS metastasis.

## 2. Bone Microenvironment and OS Metastasis

### 2.1. Mesenchymal Stem Cells and OS Metastasis

The most important factors in the bone microenvironment, which is considered to promote OS metastasis, are MSCs (Figure 1). MSCs are pluripotent stem cells and highly associated with tumor development, metastasis, and drug resistance, and they are a source of OS [25]. For example, the deficiency of TP53 and *Rb* gene, the aneuploidization and genomic loss of *P16/Cdkn2a* are common causes of the transition of MSCs to OS cells [18,26]. However, OS-derived MSCs have no characteristics of tumors but are highly similar to normal MSCs [10]. Despite all this, a growing number of researchers suggest that MSCs existing in the bone microenvironment can promote the invasion and metastasis of OS. In rat OS model, after 3 weeks and 5 weeks subcutaneous injection of rat OS COS1NR cells, the intravenous injection of MSCs had no effect on tumor growth, whereas, it promoted pulmonary metastasis significantly [27]. Meanwhile, gene expression analysis showed that the focal adhesion, cytokine–cytokine receptor and extracellular matrix–receptor pathway significantly changed in MSCs compared to COS1NR cells [27]. The pathway molecules that dramatically altered were highly related to the tumor metastasis and angiogenesis such as the CXCL12/CXCR4 axis, MMP-2, and MMP-9 [27,28,29] (Table 1). Some reports find that the interaction between MSCs and OS tumor cells is bidirectional (Figure 1). For example, tumor cells can modulate their microenvironment which, in turn, becomes more beneficial to tumor growth through metabolic reprogramming [30]. In this situation, MSCs are the modulators of OS metabolism. In addition, MSCs can secrete more lactic acid by upregulated lactate monocarboxylate transporters [30]. This process is driven by the oxidative stress induced by OS to improve the mitochondrial activity of OS and then lead to the OS metastasis [30].

Reports about tumor-derived extracellular vesicles (EVs) reveal the direct interaction of MSCs and OS cells [33,36] (Table 1). The preclinical mouse model also indicates the Es-mediated cross-talk between OS cells and MSCs [38]. The EVs carrying a membrane-associated form of TGF-β stimulate the IL-6 expression in MSCs, which is called tumor extracellular vesicle-educated mesenchymal stem cells (TEMSCs) [33]. Compared to the normal MSCs, TEMSCs can activate STAT3 expression in OS and facilitate the pulmonary metastasis [33]. Meanwhile, the activation of STAT3 also increases the drug resistance of OS [33]. MicroRNA derived from EVs, such as hsa-miR-195 and has-miR-148a, can also increases the aggressiveness and metastasis of OS through targeting MMP1 and PTK2 [36,37] (Table 1).

MSCs are driven by oxidative stress induced by OS and undergo metabolism reprogramming. Lactate production is increased which promotes the migration ability of OS cells [30]. OS cells can acidify the microenvironment where they are embedded. This highly acidic environment triggers MSCs to secrete many different types of factors, including mitogenic, clonogenic, chemotactic, and pro-migratory factors via activating the NF-κB pathway (Table 2) [31]. These factors, in turn, promote invasion and metastasis of OS. For example, IL-8 secreted by MSCs can activate the CXCR1, a member of the chemokine receptor family, and further improves p-Akt expression leading to anoikis resistance of OS cell and the advancement of pulmonary metastasis [39]. OS cells also can activate the CXCR1 through autocrine of IL-8 and then promote their own pulmonary metastasis through the Akt pathway [39]. On the other hand, the contact of MSCs and OS cells can promote the tumor invasion and migration through inducing the MAT in OS cells (Figure 1) [32]. Firstly, OS cells secrete chemokines, including MCP-1 (CCL2), TGF-β, and GRO-α (CXCL1), in the microenvironment [32]. Then MSCs are recruited into contact with OS cells, trans-differentiate into cancer-fibroblasts, and secret more MCP-1(CCL2), GRO-α (CXCL1), TGF-β, IL-6, and IL-8 in the microenvironment [32]. These cytokines active the small GTPase RhoA to induce the MAT in the OS cells finally [32] (Table 1).

### 2.2. Effect of Hypoxia and Acidosis Environment on OS Metastasis

Hypoxia and acidic condition are common features of the bone microenvironment. Aberrant expressions of a number of genes induced by hypoxia and acidic conditions promote the metastasis of OS (Table 2). Hypoxia triggers the expression of Hypoxia-inducible factor (HIF) which is the main focus of research in OS metastasis [35,41,43,44,45,46,47]. Recently, miR-20b and miR-33b were reported to directly target HIF1α [35,41]. By decreasing the expression of HIF1α, they could inhibit the invasion and proliferation of MG63 and U2OS cells [35]. HIF2α and HIF2PUT are upregulated in clinical OS samples [45]. Moreover, upregulation of HIF2α and HIF2PUT have significantly correlated with OS at the clinical stage, distant metastasis, and poor survival rate [45]. Angiogenesis is necessary in the metastasis progress. LncRNA MALAT1 is found to induce the pro-angiogenic function and is highly associated to the hypoxia responses and OS development [43]. Meanwhile MALAT1 participates in the mTOR/HIF1α loop [43]. In hypoxia microenvironment of OS, ANGPTL2 is induced in patient samples, and its expression was controlled by HIF1α [46]. Moreover, overexpression of ANGPTL4 facilitates the migration and proliferation and improves the osteoclastogenesis and bone resorption [44]. Additionally, SENP1 is upregulated and positively accommodates the HIF expression to promote the invasion and EMT process in OS cells [47].

The balance of pH in bone microenvironment is important to keep normal bone formation and bone resorption. The acidosis microenvironment promotes OS progress by activation of mesenchymal stem cells [31]. Moreover, after short time stimulation of acid, sequencing data show that the expression of stress factors, growth factors, and immunoregulatory molecules are upregulated including NFκB1, RelB, RelA, CSF3, IL-1A, IL-23A, IL-1RN, IL-6, IL-8, CXCL1,CXCL2, CXCR4, CXCL5, CCL5, CCR7 CSF2/GM-CSF, CSF3/G-CSF, BMP-2, and MMP-2. The levels of IL-6 and IL-8 are the highest in all the factors (Table 2) [31]. GM-CSF and G-CSF promote the colony formation and are related to the immunoreaction [31]. In addition, CXCL1, CXCL5, and CCL5 affect OS growth and metastasis [31].

### 2.3. Chemokines and OS Metastasis

Chemokines are composed of four subfamilies including CCL, CXCL, CX3C, and XCL [48]. The functions of chemokines are closely related to immune cells which can guide their migration [49]. Recently, increasing research show that chemokines can communicate with OS in the bone microenvironment [50,51]. Clinicopathological analysis shows that CXCR4, receptor of CXCL12, and MMP-9 are overexpressed in OS with lung metastasis patients compared to the patients without metastasis [52]. CXCL12/CXCR4/CXCR7 was activated when a co-culture of bone marrow mesenchymal stem cells and OS cells (Table 1) [53]. At the same time, the activation of CXCL12/CXCR4/CXCR7 in the co-culture model boosts the OS invasion [53]. Importantly, the CCL5/CCR5 axis not only enhances the migration of OS via αvβ3 but also facilitates tumor angiogenesis via PKC delta/c-Scr/HIF-1α pathway [54] (Table 1). The levels of IL-8 in the serum of OS patients are higher than normal patients [55]. IL-8 from OS cells autocrine or from MSCs paracrine promote OS cells invasion and pulmonary metastasis through the IL-8/CXCR1/Akt signaling pathway [34,39] (Table 1). As previously described, the acidic conditions induce various chemokines’ expression in MSCs to promote progression of OS.

### 2.4. Functions of Extracellular Vesicles in the Tumor Microenvironment

EVs in the tumor microenvironment, as a medium of communication among cells, play an important role in tumor development and metastasis. EVs can carry miRNA, cytokines, and small molecule proteins, such as TGFβ and MMP [56,57]. MSCs-derived EVs promote OS growth by activation of the Hedgehog and PI3K/AKT signal pathways [58,59]. Meanwhile, OS-derived EVs modulate the transformation of MSCs by regulating the hypomethylation of LINE-1 in MSCs [60]. Moreover, OS-derived EVs change the bone microenvironment remodeling through influenced genes’ expression [60,61]. EVs derived from highly metastatic OS clonal variants induce metastasis of poorly metastatic clones in mouse model [62]. More descriptions of EVs has been comprehensively summarized by Perut [63].

## 3. Signal Pathways in OS Metastasis

### 3.1. PI3K/Akt Signaling Pathway

PI3K/Akt is one of the most important intracellular signal transduction pathways that regulates cell motility, growth, proliferation, adhesion, and cell survival [64]. An increasing number of studies show that PI3K/Akt signaling pathway components are abnormally expressed in human cancer including OS [65]. Immunostaining analysis in primary OS cases show that PI3K/Akt signaling is highly and significantly related with poor prognosis [66]. Moreover, activation of Akt is highly implicated in lung metastasis. There are many reports showing that aberrant expression of proteins can active the PI3K/Akt signaling pathway facilitating the progression of OS [67,68,69,70,71,72,73,74,75] (Figure 2). Intracellular adhesion molecule-1 (ICAM-1), a surface glycoprotein, takes part in cell–ECM adhesion and promotes metastasis in cancers [67]. The Fractalkine/CX3CR1 axis can induce ICAM-1 expression to promote cell migration of OS [67]. The process is mediated by the PI3K/Akt/NF-κB signaling pathway. In the PI3K/Akt/NF-κB cascade, Fractalkine/CX3CR1 axis can phosphorylate Akt through PI3K, and the phosphorylation of Akt can active the NF-κB further; finally, the NF-κB as the transcription factor, facilitates the expression of ICAM-1 [67]. The IL-8/CXCR1 axis can directly activate the Akt signaling to enhance the resistance of OS to anoikis [39]. The apoptosis-related imprinted gene, tumor-suppressing STF cDNA 3 (TSSC3) and a prognostic marker for OS patients, can induce the autophagy in OS and inhibit the cell migration and invasion in vitro and in vivo via depressing the src-dependent PI3K/Akt/mTOR signaling pathway [68]. Overexpression of the Fibulin-4, an extracellular matrix protein playing a key role in stabilizing the matrix structure, can also active the PI3K/Akt/mTOR signaling pathway to promote OS cell invasion and metastasis [69]. Meanwhile, EMT, an important cellular process in cancer cell metastasis, is accelerated with the upregulation of Fibulin-4 [69]. The PI3K/Akt/mTOR signaling pathway is also active via the high expression of GPNMB and promotes OS cell metastasis and proliferation [70]. Transcription factors also take effect on the activation of PI3K/Akt such as overexpression of zinc finger transcription factor ZIC2, which can activate PI3K/Akt and promote the viability, migration, and invasion of OS cells [71].

Non-coding RNA is also an important regulator in activation of the PI3K/Akt pathway. Silence of LncRNA H19 and LINC00968 can decrease the activation of the PI3K/Akt/mTOR signaling pathway in vitro [72]. Overexpression of LINC00628 obviously suppresses the related protein expression level of the PI3K/Akt signaling pathway [73]. The LncDANCR via competitive combination of miR-33a-5p promotes the expression of AXL and then facilitates the invasion and metastasis of OS in vitro and in vivo through the PI3K/Akt signaling pathway [74]. Moreover, the PI3K/Akt signaling pathway has the potential as a therapy target. For example, Celastrol can inhibit OS cell line metastasis via inactivation of the PI3K/Akt/NF-κB signaling pathway in vitro [75].

### 3.2. Wnt/β-Catenin Signaling Pathway

There are three different Wnt signaling pathways: the canonical Wnt pathway, the noncanonical planar cell polarity pathway, and the non-canonical Wnt/calcium pathway [76]. The canonical Wnt signaling pathway is the Wnt/β-catenin pathway including many molecules such as TCF/LEF, APC, Axin, GSK-3β, CK1α, and LPR5/6. The activation of the Wnt signaling pathway is frequently detected in most cancers like OS [77]. Many different molecules can enhance OS’s malignancy through activating the Wnt/β-catenin signaling pathway.

Non-coding RNAs have been star molecules in recent reports. Overexpression of miR-135b, an oncogenic miR in OS, can promote OS invasion and metastasis in vitro and in vivo through activating the Wnt/β-catenin signaling pathway via directly targeting GSK-3β, APC, β-TrCP, and CK1α [78]. The Notch signaling pathway also can be activated by Wnt/β-catenin-dependent and -independent mechanisms in OS cells simultaneously [78]. Meanwhile inhibition of miR-135b not only decreases the activation of the Wnt/β-catenin and Notch signaling pathway but also depresses OS metastasis and reduces recurrence in OS xenografts models [78]. Similarly, the Wnt/β-catenin signaling pathway is activated directly or indirectly by miR-183, miR-184, miR-146b-5p, etc., (Table 3) to promote OS metastasis and invasion [79,80,81,82,83,84,85,86]. Meanwhile, the activation of Wnt/β-catenin can increase the expression of Runx2 to facilitate the metastasic-related genes expression in OS [87].

Wnt/β-catenin is related to EMT. The signaling cascade of Wnt/β-catenin induced by tumor-suppressing STF cDNA3 downsregulation results in the accumulation of the β-catenin and snail, finally enhancing the upregulation of Wnt target genes and mesenchymal genes to promote EMT [88]. The Wnt antagonist, APCDDI, is downregulated by methylation in the promoter to activate the Wnt/β-catenin signaling pathway elevating the ability of invasion and migration of OS cells in vitro and in vivo [89].

In summary, the Wnt/β-catenin signaling pathway is widely activated in OS and highly related to the invasion and metastasis of OS. Therefore, an increasing number of studies focus on the Wnt/β-catenin signaling pathway, and it would be a suitable therapy target in the future treatment of OS.

### 3.3. MAPK/ERK Signaling Pathway

MAPK/ERK is a complex cascade of signaling pathway that transfers signals from a receptor on a cell’s surface to the DNA in a cell’s nucleus, involving various cellular processes such as cell motility, survival, apoptosis, differentiation, and proliferation [89]. The MAPK/ERK signaling pathway includes many components such as ERK1/2/3/4/5, JNK1/2/3, SAPK, and P38 [89]. Any mutation in the components of this pathway can result in the stop or start of the signaling and along with tumorigenesis [90]. Paris saponin VII (PS VII), Delphinidin can effectively inhibit the activation of the MAPK/ERK signaling pathway and further suppress the OS invasion, migration and promote apoptosis, which indicates that MAPK/ERK signaling pathway is activated in OS [91,92]. Over expressions of MAP2K6, MAP4K3, and DUSP1 are correlated with poor clinical prognoses [93]. Inversely, patients with decreasing expression of MAP4K3 can achieve a better treatment effect [93]. Moreover, the expression of DUSP1, one of the protein phosphatases which inhibits MAPKs through dephosphorylation, is significantly increased in metastasis patients and post-chemotherapy patients, suggesting that DUSP1 gene is related to OS metastasis and drug resistance [93]. Continuously, there are some other cellular molecules, such as ONZIN, macrophage migration inhibitory factor (MIF) activates the MAPK/ERK signaling pathway via directly or indirectly phosphorylating ERK1/2 to promoting OS invasion and metastasis [94,95]. For example, MIF expression is increased both in serum and OS tissues of patients along with poor survival rate and highly lung metastasis rate [95]. MIF promotes metastasis and proliferation of OS cells by activating the RAS/MAPK signaling pathway via enhancing p-ERK1/2, p-SRC, p-MEK1/2, and upregulation of RAS-GTP [95]. Moreover, silence of MIF decreases the p-ERK level and inhibits the ability of metastasis and proliferation in vitro and suppresses OS lung metastasis in vivo and raises the sensitivity of OS cells to chemotherapy drugs [95]. Higher levels of ERK1/2 and STAT3 has been confirmed to be associated with poor prognosis of OS [96]. STATs, as members of the JAK–STAT signaling pathway, are considered as oncogenes in OS [97]. The downregulation of the JAK–STAT signaling pathway by miR-125b and miR-126 suppresses migration, invasion, and proliferation in OS cell lines [98,99].

### 3.4. Hedgehog Signaling Pathway

The Hedgehog (Hh) signaling pathway, composed of twelve-transmembrane Patched protein (PTc), seven-transmembrane protein smoothened (SMO), and glioma-associated oncogene homologs (GLI), is a conserved signaling pathway participating in cell migration, differentiation, growth, and polarity [100]. The function of Hh signaling in OS metastasis has been reviewed explicitly elsewhere [101]. Briefly, the Hh pathway is activated broadly in many different human cancers including OS [101,102]. The activation of the Hh pathway mediated by various agents facilitates OS invasion and metastasis [101]. Ribosomal protein S3 (RPS3), Smad, Yes-associated protein 1, and LncRNA H19 modulates the Hh/Gli pathway to improve the OS invasion and metastasis [103,104,105]. Some data show that the Smo antagonist and Degalactotigonin suppresses OS progression by inactivation of the Hh/Gli pathway [106,107]. Moreover, the Hh/Gli pathway can promote OS metastasis by interacting with other signaling pathways such as the PI3K/AKT pathway and the Wnt pathway [106,107]. Additionally, details of the treatment strategies targeting the Hh signaling pathway have been review by Yaoet et al. [101].

### 3.5. Notch Signaling Pathway

The Notch signaling pathway is a conserved pathway modulated cell–cell communication, which is important for regulating cell processes [108]. The Notch proteins are transferred on the cell surface to modulate cellular signal transmission. Many data show that the Notch pathway is positive in various cancers including OS [109]. The Notch pathway is activated with the decrease in DTX1 [110]. At the same time, the depletion of DTX1 results in the improvement of Notch1 in OS cells [110]. Moreover, the activation and recycling rate of Notch is suppressed by the activation of PI5P4Kγ [110]. Some other reports show that LncRNAs and microRNAs play important roles in activating the Notch pathway. The Notch pathway is activated through the upregulation of Notch2 which modulates LncRNA SNHG12 sponging miR-195-5p in OS [111]. Moreover, SNHG12 is highly expressed in OS tissues and cell lines and is significantly related with the poor prognosis in OS patients [111]. The motilities of OS cells are suppressed by the silence of SNHG12 [111]. Lnc CRNDE is significantly highly expressed in OS samples from patients, particularly in metastasis patients [112]. Overexpression of CRNDE promotes the upregulation of Notch1 activating the Notch pathway and induces the EMT progress in OS cells [112]. In other words, the Notch pathway may have a close relationship with EMT in OS metastasis. Clinical data and animal model suggest that miR-135b, similar to oncogenic genes, improves pulmonary metastasis, cancer cell stemness, and tumor recurrence in OS [78]. Meanwhile, miR-135b activates the Notch pathway and Wnt/β-catenin pathway through inhibiting the negative modulator of these pathway in cancer stem cells of OS [78]. For example, miR-135b targets GSK-3β, APC, β-TrCP, and CK1α in Wnt/β-catenin pathway and inhibits the TET3 in Notch pathway [78]. The combined treatment of chemotherapy and natural compounds (traditional Chinese medicine) reveals that the Notch pathway is a therapeutic target in OS metastasis and development [113].

## 4. The Bone Microenvironment as a Promising Treatment

The treatment of OS, usually conducted with the combination of chemotherapy and surgery, has not been significantly improved in recent decades. The development of OS is closely related to the bone microenvironment. There may be a new therapeutic strategy that uses the bone microenvironment as target. Although targeting a certain type of cell is impractical due to the heterogeneity of OS, improving the physical conditions of the bone microenvironment may achieve a certain therapeutic effect. Tumor hypoxia is one of the main reasons for treatment failure. TH-302, as a developed hypoxia-activated DNA cross-linking pro-drug, shows potent hypoxia-dependent cytotoxicity. TH-302, combined with doxorubicin, reduced lung metastases of OS [114]. Acidic conditions in the tumor microenvironment increased tumor resistance. Doxorubicin cytotoxicity, compared with the standard pH7.4, was reduced in pH6.5 condition. Combining omeprazole, a proton pump inhibitor targeting lysosomal acidity, with doxorubicin significantly reduce tumor volume and body weight loss by improving the doxorubicin cytotoxicity [115]. As described above, EVs, as a messenger of communication among cells in the microenvironment, promotes the development and metastasis of OS. To prevent the production of EVs or directly target the EVs in tumor microenvironment may be also an effective strategy to treat OS. However, current research on EVs is mainly focused on using it as a drug or delivery system [116].

## 5. Conclusions and Perspectives

Previous studies showed that the bone microenvironment promotes the OS metastasis or malignancy. In this review, we briefly reviewed the OS and bone microenvironment; mainly discussed the function of mesenchymal stem cells, acidosis, hypoxia, and chemokines in the microenvironment to promote the OS migration, invasion, and lung metastasis; and extensively summarized the signaling pathways activated in the OS that facilitate the metastasis of OS including the PI3K/Akt, Wnt/β-Catenin, MAPK/ERK, Hedgehog, and Notch signaling pathways. We found that the function of mesenchymal stem cells and chemokines are hot spots in the research of OS and also are the most important factors in the bone microenvironment to promote the progress of OS. There are always some cross-talks between different pathways in tumor occurrence and development, which provides a comprehensive understanding of tumor occurrence and development. Furthermore, the IL-8/CXCR1 axis and CXCL12/CXCR4/CXCR7 axis are commonly activated in the bone microenvironment when OS metastasis occur [29,31,33,39,53]. IL-8 and CXCL12, as the secreted molecules, are detected easily through peripheral blood. IL-8 and CXCL12 may be biomarkers of OS metastasis and IL-8/CXCR1 axis and CXCL12/CXCR4/CXCR7 axis may be as the potential therapeutic targets for OS metastasis in the future.

## Figures and Tables

**Figure 1 ijms-21-06985-f001:**
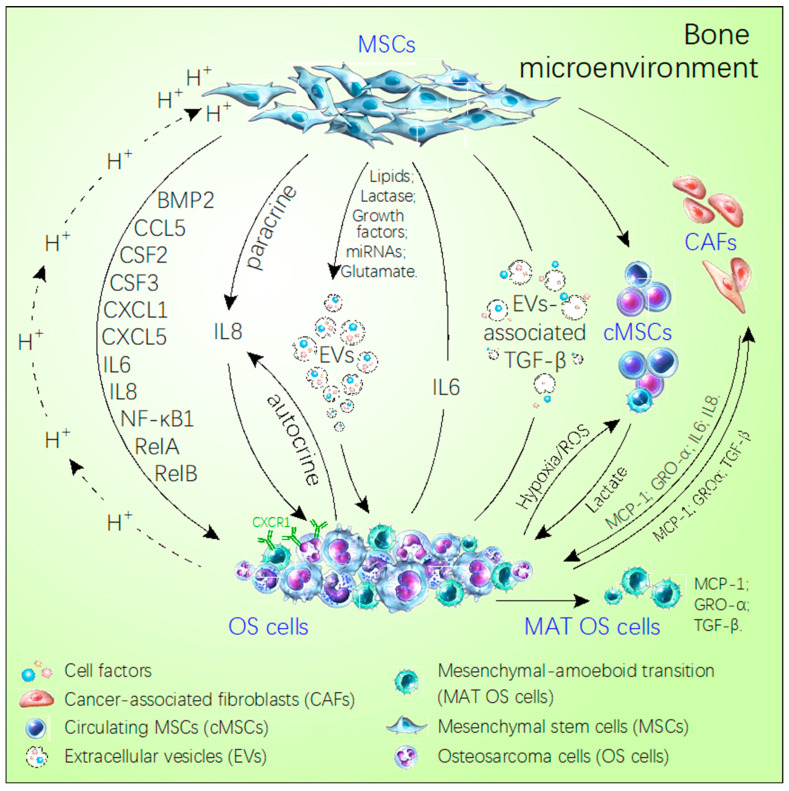
Schematic diagram of the cross-talk between MSCs and OS cells in the bone microenvironment. MSCs directly secrete some factors or choose the extracellular vesicles as the carrier to transport miRNAs, growth factor, lipids, glutamate, lactase, which promote OS metastasis. Meanwhile, OS utilizes the acidic environment, hypoxia and extracellular vesicles inducing the MSCs secreting cell factors to facilitate their own growth and metastasis, like IL-8, IL-6, RelA, RelB, NF-κB1, CSF2/GM-CSF, CSF3/G-CSF, BMP2, CCL5, CXCL5, CXCL1. In addition, OS via secreted MCP-1, GRO-α, and TGF-β induces MSCs trans-differentiating the cancer-associated stem cells to express more MCP-1. GRO-α, IL-6, and IL-8 promote the MAT of OS. The solid arrows refer to the direction of the cell or factor from the MSCs. The dotted arrows refer to the direction of cells and factors from OS.

**Figure 2 ijms-21-06985-f002:**
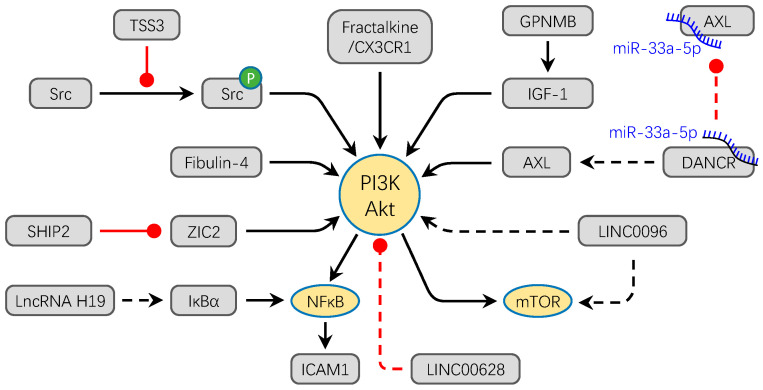
The PI3K/Akt signaling pathway converges to facilitate the progression OS. Most factors directly activate PI3K/Akt signaling pathway to promote the migration and invasion of OS. For example, overexpression of Fibulin-4, Fractalkine/CX3CR1, ZIC2, LINC0096, and lncRNA H19 promote the osteosarcoma metastasis through the activation of PI3K/Akt signal pathway. Meanwhile, GPNMB and DANCR/miR-33a-5p activate the PI3K/Akt pathway by promoting the expression of IGF-1 and AX, respectively. TSSC3 inhibits the phosphorylation of Src to activate the PI3K/Akt signaling pathway. Conversely, overexpression of LINC00628 inhibits the phosphorylation of PI3K and Akt. The solid lines refer to protein molecules and the dotted lines refer to non-coding RNAs in this figure. Black arrows show the promotion of molecules, while red dots show the inhibition of molecules.

**Table 1 ijms-21-06985-t001:** Chemokines and other factors that affect OS metastasis in the bone microenvironment.

Factor	Source	Pathway	Target	In Vitro or In Vivo	References
CXCL1 (GRO-α)	MSCs	NF-κB pathway in MSCs	OS cells	In vitro	[31]
	OS cells	Trans-differentiate into cancer-associated fibroblasts in MSCs	MSCs	In vitro	[32]
	cancer-associated fibroblasts	MAT in OS cells	OS cells	In vitro	[32]
CXCL5	MSCs	NF-κB pathway in MSCs	OS cells	In vitro	[31]
IL-6	MSCs	IL-6/STAT3 signaling pathway in OS cells	OS cells	In vitro and in vivo	[33]
	MSCs	NF-κB pathway in MSCs	OS cells	In vitro	[31]
	cancer-associated fibroblasts	MAT in OS cells	OS cells	In vitro	[32]
IL-8	MSCs	NF-κB pathway in MSCs	OS cells	In vitro	[31]
	MSCs and OS cells	IL-8/CXCR1/Akt signaling pathway, MAT	OS cells	In vitro and in vivo	[34]
	MSCs, cancer-associated fibroblasts	MAT in OS cells	OS cells	In vitro	[32]
CXCL12	MSCs	CXCL12/CXCR4/CXCR7	OS cells	In vitro	[27,35]
CCL2 (MCP-1)	OS cells, cancer-associated fibroblasts	MAT in OS cells	OS cells	In vitro	[32]
CCL5	MSCs	NF-κB pathway in MSCs	OS cells	In vitro	[31]
Lactate	MSCs	metabolic reprogramming OS cells	OS cells	In vitro	[30]
BMP-2	MSCs	NF-κB pathway in MSCs	OS cells	In vitro	[31]
NF-κB1, RelA, RelB	MSCs	NF-κB pathway in MSCs	OS cells	In vitro	[31]
CSF2/GM-CSF	MSCs	NF-κB pathway in MSCs	OS cells	In vitro	[31]
CSF3/G-CSF	MSCs	NF-κB pathway in MSCs	OS cells	In vitro	[31]
TGF-β	OS extracellular vesicle	IL-6/STAT3 signaling pathway in OS	MSCs	In vitro and in vivo	[32,33]
has-mir-195	MSCs extracellular vesicle	FAK/PTK2 in OS	OS cells	In vitro	[36]
has-mir-124	MSCs extracellular vesicle	Has-mir-124/Rac1	OS cells	In vitro	[36,37]

**Table 2 ijms-21-06985-t002:** The hypoxia and acidosis in the bone microenvironment affect OS metastasis (↑: Up-regulation).

Physical Element	Factors	Interacting Molecule	Efficacy	Reference
Hypoxia	HIF-1α	CXCR4	migration ↑	[37]
		TGF-β	osteolytic bone metastases ↑	[40]
		miR-20b	invasion and proliferation ↑	[35]
		miR-33b		[41]
		BMPR2	distant metastasis and poor survival rate ↑	[42]
		LncRNA MALAT1	pro-angiogenic ↑	[43]
		ANGPTL4	migration, proliferation ↑	[44]
	HIF2PUT	HIF2α	distant metastasis ↑	[45]
Acidic condition	CXCL1		growth, metastasis ↑	[31]
	CXCL2			
	CXCL5			
	CXCR4			
	BMP2		colony formation ↑	[31]
	CSF2/GM-CSF			
	CSF3/G-CSF			
	IL1A			
	IL1RN			
	IL23A			
	IL-6			
	IL-8			
	MMP2			
	NFκB1			
	RelA			
	RelB			

**Table 3 ijms-21-06985-t003:** Non-coding RNAs related with Wnt/β-catenin pathway promote the OS metastasis and invasion (↑: Up-regulation).

Non-coding RNA	Target Molecule	Effect	In Vitro or In Vivo	Reference
miR-135b,	GSK3β, CK1a, TET3	lung metastasis, tumor recurrence ↑	In vitro or in vivo	[78]
miR-183	LRP6	migration, invasion ↑	In vitro	[79]
miR-146b-5p	ZNRF3	invasion, metastasis, chemoresistance ↑	In vitro	[80]
miR-26a	GSK-3β	proliferation, migration, invasion ↑	In vitro	[82]
miR-214	β-catenin	proliferation ↑	In vitro	[83]
miR-342-3p	AEG-1	proliferation, migration, invasion ↑	In vitro	[81]
LncSox4	β-catenin	cell viability ↑	In vitro	[84]
Lnc-SNHG1	miR-557/WNT2B	migration, EMT process, tumor growth ↑	In vitro or in vivo	[85]

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
