# Peer review of "Bone Microenvironment and Osteosarcoma Metastasis"

_ijms, 2020, doi:10.3390/ijms21196985_

Round 1
Reviewer 1 Report
The paper drafted by Yang et al and entitled 'Bone microenvironment and osteosarcoma metastasis' is trying to review the cross-talk between bone microenvironment and osteosarcoma tumor cells. The paper is well written and can be understood well, it however still needs some modification and priming as follows: Major points: The authors need to elaborate more on MSC and Bone Marrow elements. MSC is a pool of different cells, the same is true for Bone marrow. Within a few sentences, authors could expand it in more detail. It would be very helpful if the authors could discuss a bit more about the cellular origin of OS in the introduction. The prominent gene alteration, mutations etc that are involved in OS should be explained in detail. The same is true about the sites of metastasizing. As the extracellular vesicles are getting more attention and becoming more important in terms of cell signaling it would be very helpful if authors could expand this part and even draw a cartoon for it. Minor points: This paper: https://doi.org/10.1016/j.cyto.2016.06.017 is very related and needs to be cited specifically in the section of stat3. The following article studied extensively the role of microenvironment on metastasis fro other pints of view and would be good to cite them as well: https://doi.org/10.1038/s41568-019-0165-1 https://doi.org/10.1186/s12964-020-0530-4
Author Response
Point 1: The authors need to elaborate more on MSC and Bone Marrow elements. MSC is a pool of different cells, the same is true for Bone marrow. Within a few sentences, authors could expand it in more detail.
Thank you for your advice. There have been many articles showing MSC and bone marrow elements, it is why there is no detailed discussion. We will cite these articels in the revised manuscript in case the reader would like to have further knowledge about MSCs. And in order to have a clear understanding, we will modify the manuscript at the beginning of the article to expand it. see below
Section 1, first paragraph, Line 1 to line 5 “Bone microenvironment is comprised of bone marrow and mineralized extracellular matrix. Bone marrow contains two different cell types, one is hematopoietic stem cells with hematopoietic function, and the other is bone marrow mesenchymal stem cells that are responsible for differentiating into non-blood cells components in bone, including osteoblasts, osteoclasts, osteocytes, fibroblasts, adipocytes ect.”
We will incorporated this description in the revised manuscript.
Point 2: It would be very helpful if the authors could discuss a bit more about the cellular origin of OS in the introduction.
Unfortunately, the cellular origin of OS is still unclear. We will discuss it in this article section 1 and focus on the transformation of osteoblast and MSCs which is the two most studied areas.
Section 1 third paragraph, line 9 to line 11 For example, some papers reported that the deletion of TP53 and Rb can cause OS transformation of osteoblast. The loss of Rb can trigger the transformation of MSCs into OS, and the overexpression of C-myc has the similar consequence in MSCs.
Point 3: The prominent gene alteration, mutations etc that are involved in OS should be explained in detail. The same is true about the sites of metastasizing.
Thank you for your advice. There are many papers reported the genes alteration, mutation and aberrant expression in detail, such as doi:10.1038/labinvest.3780431, doi:10.1155/2011/959248, doi:10.7314/APJCP.2014.15.15.5967, doi:10.1089/dna.2006.0505,Molecular Pathogenesis of Osteosarcoma.
We will cited these articles in section 1, and add some necessary explanation in the revised manuscript. Section 1, third paragraph, line 11 to line 16
Point 4: As the extracellular vesicles are getting more attention and becoming more important in terms of cell signaling it would be very helpful if authors could expand this part and even draw a cartoon for it.
Thank you for your valuable suggestions. we expanded the EVs in section 2, but didn’t draw cartoon. Because EVs in OS was summarized very clearly in doi:10.3389./fonc.2019.01342.
2.4 Functions of Extracellular Vesicles in Tumor Microenvironment
EVs in the tumor microenvironment, as medium of communication between cells, play an important role in tumor development and metastasis. EVs can carry miRNA, cytokines and small molecule proteins, such as TGFβ, MMP. MSCs-derived EVs promote OS growth by activation of Hedgehog and PI3K/AKT signal pathways. Meanwhile, OS-derived EVs modulate the transformation of MSCs by regulating the hypomethylation of LINE-1 in MSCs. Moreover, OS-derived EVs change the bone microenvironment remodeling through influenced genes expression. EVs derived from highly metastasis OS clonal variants induce metastasis of poorly metastasis clones in mouse model. More description of EVs has been comprehensively summarized by Perut, F.
Point 5: This paper: https://doi.org/10.1016/j.cyto.2016.06.017 is very related and needs to be cited specifically in the section of stat3.
Thank you for your advice. We cites this paper: https://doi.org/10.1016/j.cyto.2016.06.017 section 3 MAPK/ERK signaling pathway in the revised version.
Point 6:The following article studied extensively the role of microenvironment on metastasis from other pints of view and would be good to cite them as well: https://doi.org/10.1038/s41568-019-0165-1 https://doi.org/10.1186/s12964-020-0530-4
Thank you for your advice, we cited the two papers in section 1. This is very helpful for our paper. References 6,9
Reviewer 2 Report
The authors have provided a comprehensive review of various microenvironmental factors that influence metastasis of osteosarcoma cells. However, the manuscript needs to be extensively edited by a native English speaker.
Section 2.1: Please change p53 to TP53 and italicize. Similarly use the correct nomeclature for Cdkn2.
Also please include the frequency of these alterations from publicly available sequencing databases such as TCGA and GENIE.
In this section the authors state that "For example, tumor cells can modulate their microenvironment that in turn, becomes more beneficial to tumor growth through metabolic reprogramming"
This part is quite confusing. At first the authors mention of reciprocal interaction between the tumor and its microenvironment that favors tumor growth. But then they go on to talk about MSC mediate metabolic reprogramming without describing how the tumor affects its microenvironment.
The authors should consider adding some information on therapeutic intervention strategies to prevent OS metastasis potentiated by MSC, hypoxia, acidosis, cell signaling, etc
Author Response
Point 1: Section 2.1: Please change p53 to TP53 and italicize. Similarly use the correct nomenclature for Cdkn2.Also please include the frequency of these alterations from publicly available sequencing databases such as TCGA and GENIE.
Thank you for your advice. We changed p53 to TP53 and italicize, also correct Cdkn2. we are sorry about the wrong nomenclature of Cdkn2. The right nomenclature of this gene is Cdkn2a or P16.
Point 2: In this section the authors state that "For example, tumor cells can modulate their microenvironment that in turn, becomes more beneficial to tumor growth through metabolic reprogramming" This part is quite confusing. At first the authors mention of reciprocal interaction between the tumor and its microenvironment that favors tumor growth. But then they go on to talk about MSC mediate metabolic reprogramming without describing how the tumor affects its microenvironment.
Thank you for your advice. We have revised this part according to your suggestion.
Section 2.1 third paragraph, line 1 MSCs are driven by oxidative stress that is induced by OS to undergo metabolism reprogramming, the lactate production is increased which promotes the migration ability of OS cells.
Point 3: The authors should consider adding some information on therapeutic intervention strategies to prevent OS metastasis potentiated by MSC, hypoxia, acidosis, cell signaling, etc
Thank you for your advice. We added some information about the therapeutic intervention strategies in section 4.
Round 2
Reviewer 1 Report
Accept.